# Differential Gene Expression Correlates with Behavioural Polymorphism during Collective Behaviour in Cockroaches

**DOI:** 10.3390/ani12182354

**Published:** 2022-09-08

**Authors:** Isaac Planas-Sitjà, Jean-Louis Deneubourg, Denis L. J. Lafontaine, Ludivine Wacheul, Adam L. Cronin

**Affiliations:** 1Department of Biology, Tokyo Metropolitan University, 1-1 Minami-Osawa, Hachioji, Tokyo 192-0397, Japan; 2Center for Nonlinear Phenomena and Complex Systems (CENOLI)-CP 231, Faculty of Sciences, Campus La Plaine NO.5, Université Libre de Bruxelles, Boulevard du Triomphe, 1050 Bruxelles, Belgium; 3RNA Molecular Biology, Faculty of Sciences, Biopark Campus, Université Libre de Bruxelles, Fonds de la Recherche Scientifique (F.R.S./FNRS), 6041 Gosselies, Belgium

**Keywords:** collective behaviour, gene expression, aggregation, animal personality, insects, cockroaches

## Abstract

**Simple Summary:**

It is currently well accepted that animals differ from one another in their behaviour and tendency to perform actions, a property we refer to as animal personality. In group-living animals, variation in animal personality can be important to determine group survival, as it determines how individuals interact with each other and with their environment. However, we have little knowledge of the proximal mechanisms underlying personality, particularly in group-living organisms. Here, we investigate the relationship between gene expression and two behavioural types (bold and shy) in a gregarious species: the American cockroach. Our results show that bold individuals have upregulated genes with functions associated with sensory activity (phototaxis and odour detection) and aggressive/dominant behaviour, and suggest that social context can modulate gene expression related to bold/shy characteristics. This work could help identify genes important in the earliest stages of group living and social life, and provides a first step toward establishing cockroaches as a focal group for the study of the evolution of sociality.

**Abstract:**

Consistent inter-individual variation in the propensity to perform different tasks (animal personality) can contribute significantly to the success of group-living organisms. The distribution of different personalities in a group influences collective actions and therefore how these organisms interact with their environment. However, we have little understanding of the proximate mechanisms underlying animal personality in animal groups, and research on this theme has often been biased towards organisms with advanced social systems. The goal of this study is to investigate the mechanistic basis for personality variation during collective behaviour in a species with rudimentary societies: the American cockroach. We thus use an approach which combines experimental classification of individuals into behavioural phenotypes (‘bold’ and ‘shy’ individuals) with comparative gene expression. Our analyses reveal differences in gene expression between behavioural phenotypes and suggest that social context may modulate gene expression related to bold/shy characteristics. We also discuss how cockroaches could be a valuable model for the study of genetic mechanisms underlying the early steps in the evolution of social behaviour and social complexity. This study provides a first step towards a better understanding of the molecular mechanisms associated with differences in boldness and behavioural plasticity in these organisms.

## 1. Introduction

Many animals exhibit consistent differences in behavioural propensity over time and context, a property which has been variously termed animal personality, behavioural syndromes, idiosyncrasy or polyethism [1,2]. Animal personality can have important evolutionary and ecological implications, as behaviour determines how organisms interact with their environment [3,4,5]. In addition to effects at the individual level, personality in group-living organisms has been shown to be important in governing collective actions. Recent theoretical and empirical studies suggest that various collective actions in animals can be facilitated by heterogeneity among group members [6,7,8]. For instance, increasing behavioural variation among individuals in honeybee colonies improves colony efficiency and productivity [9]. However, we have little understanding of the proximate mechanisms underlying personality variation, and if a similar mechanistic basis might be found across the marked diversity of group-living organisms [10,11].

The basis of differences in behavioural propensity between individuals can be genetic, environmental or a combination of these, but in all cases should be reflected in differences in gene expression [12,13,14]. For instance, in the ant *Temonothorax longispinosus*, differences among workers in behavioural responsiveness to parasites are linked to differences in brain gene expression [15], and in female mice, differential gene expression plays a key role in sexual, parental and aggressive behaviour [16]. Research with honeybees has shown that shifts in brain gene expression can precede shifts in behavioural state, even in genotypically identical individuals [17]. Differential gene expression is thus a powerful tool to better understand the proximate mechanisms underlying animal personality, and, in turn, how this can affect the collective actions of group-living animals.

Studies on social insects have identified gene expression as a potential regulator of key elements determining group function, such as differences in social behaviour or caste determination [10,13,18,19,20,21,22]. However, most of these studies have focussed on obligately social organisms, which by definition possess complex and highly derived social behaviour. To better understand how changes in gene expression may contribute to enhancements in group performance at the early stages of social evolution, we need data from solitary and subsocial (or incipiently social) organisms [18,19,22,23]. Some recent studies on facultatively social or socially polymorphic species indicate that there may be common genetic mechanisms underlying social behaviour in different taxa [14,18,23,24,25]. Taking this one step further, studies of subsocial organisms could thus help elucidate the fundamental molecular underpinnings of collective behaviour.

Cockroaches lack the complex societies of ants, bees and termites, though the latter evolved from cockroaches [26], and have emerged as a model system for studies of gregarious behaviour and collective decision-making [27,28,29]. Domiciliary cockroaches explore the surroundings solitarily during the night but aggregate in a sheltered place during the day, usually with high fidelity for the same location [30]. Previous studies have shown that cockroaches are capable of performing collective decisions when facing a choice among several shelters [31,32]. This collective choice arises from interactions between individuals, as they are attracted to hydrocarbons on the body surface (i.e., individuals actively seek close proximity with one another), and use the presence of other individuals under shelters as a social cue to indirectly estimate shelter quality [29,33,34]. This aggregation process is important for protection, reduction of water loss and increase in growth rate [33]. Several studies have shown that cockroaches also display measurable behavioural variability in social cohesion [28], boldness [35], learning [36], behavioural plasticity [37] and thigmotaxis [38]. In the American cockroach (*Periplaneta americana*), individuals exhibit differences in the propensity to seek shelter, with some individuals settling faster and remaining longer within shelters than others [27,39]. These differences persist over time and contexts [37], indicating that this can be considered a personality trait. Faster sheltering individuals can play a key role in aggregation dynamics [40], leading to increased speed of aggregation and the number of aggregated individuals [39]. Whether these differences originate from differences in locomotory activity, sociality or photophobic response and whether they are genetically or socially induced is unknown. Our aim in this study is to use this organism to elucidate the molecular basis for sheltering propensity, a behaviour fundamental to group function and collective decision-making, thus linking individual gene expression to collective behaviour.

## 2. Materials and Methods

### 2.1. Behavioural Experiments

Experiments were carried out on 29 September 2015 at the Université libre de Bruxelles (ULB) in an experimental arena used to study collective decision-making (see Appendix A). We used adult males of *P. americana* taken from the ULB breeding facilities. The source population of cockroaches has been maintained at ULB as a breeding colony for over 20 years and individuals are thus likely to exhibit low genetic variability. Only males were used to avoid the effect of female sexual pheromones on the aggregation process [33]. It was not possible to precisely age the individuals, but as individuals accumulate damage over time due to agonistic behaviour, and we selected males which lacked any visible external damage, we can infer that they were young adults (±6 months). The setup consisted of a circular arena (100 cm diameter) covered with a paper floor layer (120 g/m^2^), surrounded by a black polyethylene ring (diameter: 100 cm, height: 20 cm); the inner surface of this ring was covered by an electric fence to prevent cockroaches from escaping [39]. A lighting source consisting of four Philips Ambiance Pro 20 W lamp bulbs provided homogeneous illumination intensity. Two shelters made of transparent Plexiglas discs (diameter: 15 cm) were placed on the arena and covered by a red filter film (Rosco E-Colour 19: fire), creating low luminosity zones. These are perceived as rest sites for cockroaches as they are photophobic and highly thigmotactic [33]. The centre of each disc was located 23 cm from the edge of the arena and 3 cm above the arena floor. Each shelter was large enough to potentially contain the entire group. In order to detect when individual insects were in the shelters, cockroaches were tagged with RFID chips (diameter: 7.1 ± 0.2 mm and weight: 107 ± 3 mg; Space-code) and a circular RFID reader was located below each shelter (under setup floor) to measure the individual time spent under each shelter. The entire setup was surrounded by white curtains to avoid spatial cues (see Appendix A for more details).

A total of 6 groups of 16 males were kept in total darkness for 48 h in Plexiglas boxes (36 × 24 × 14 cm) containing a cardboard shelter, humidified cotton wool and ad libitum food. Afterwards, a cardboard shelter containing the 16 cockroaches was introduced to the centre of the arena (with lights already turned on) and opened to let cockroaches explore the arena. We started the experiment when all cockroaches left the cardboard shelter (less than 10 s elapsed) and monitored their behaviour for 3 h. After that time, cockroaches were removed and kept in the same Plexiglas box. The six experiments were performed on the same day in two sets (9 a.m.–12 a.m. and 1 p.m.–4 p.m.), with three experiments run simultaneously in three identical arenas in each set. Past studies with cockroaches in similar setups have shown no difference in activity patterns for these time periods [30,37]. After each experiment, we quantified the total time spent under the shelters for each cockroach (individual resting time or IRT; Figure 1A). Resting time is a good predictor of individual differences in refuge use in insects [28,35,41,42,43,44]. Following aggregation experiments, we quantified and ranked individuals’ IRT. We then classified the five individuals from each group that spent the longest time sheltered as long resting time (LRT) individuals, and the five individuals from each group that spent the shortest time under shelters as short resting time (SRT) individuals. The RNA of LRT individuals (see extraction procedure below) was then pooled into three LRT samples by combining individuals from groups 1 and 2, 3 and 4, and 5 and 6, respectively. The same was done for SRT individuals, thus composing three LRT and SRT pools of 10 individuals each. If an individual did not visit any shelter during the experiment, we did not use it as it could be a sign of disease (zero time under shelters is often associated with low mobility, avoidance of congeners and low survival time after experiments; pers. obs.), and we instead took the next ranked individual within its group. In group 4, two SRT individuals were injured before removing their heads and thus were discarded. To replace them, we took the next ranked SRT individual from the same group and the next ranked SRT individual from group 5, as we always kept the 6th individual with the longest/shortest time of each group as a backup. One LRT individual from group 5 was placed by mistake in pool 1 (instead of pool 3), so we filled pool 3 with the corresponding LRT individual from group 2. We thus ended up with three pools composed of 10 LRT individuals each, and three pools composed of 10 SRT individuals each, comprising a final experimental set of 3 LRT and 3 SRT samples. In line with the terminology used in other studies of animal personality, LRT individuals could be considered shy or social, while SRT may be considered bold or asocial [45,46]. Previous work using cockroaches from the same breeding facility, in the same setup, and partly during the same year [27,37,38,39,40], has shown that individual differences in total sheltering time are consistent over time and context, and thus although we only conducted a single behavioural test, this can be considered representative of longer-term personality differences between individuals. Cockroaches from the final SRT and LRT samples were submerged in liquid nitrogen at the end of the day of the experiments (and after analysis of resting time). The head was then removed and stored at −80 °C until RNA extraction.

### 2.2. Sample Preparation and RNA Sequencing

In preparation for sequencing, in March 2016, heads without antennae were submerged again into liquid nitrogen, crushed to powder with a plastic rod and had total RNA extracted with TRIzol (following the standard protocol for RNA extraction). We used the head as it contains the main sensory organs and the brain, the control centre of behaviour. The RNA extraction was done by a person blind to the phenotype (SRT/LRT) to avoid bias during the extraction process.

Transcriptome sequencing of the six (three LRT and three SRT) samples was realized by GENEWIZ (London, UK) in November 2016. The RNA library was prepared via Poly(A) selection from total RNA. Sequencing was performed on an Illumina HiSeq2500 platform in a 2 × 100 bp paired-end configuration in high output mode (V4 chemistry) with a total of at least 250 million reads per lane, evenly distributed among samples (obtaining around 9000 Mbp per sample). Raw reads were trimmed with CLC Genomics Server 8.0 (error rate < 0.01). A de novo transcriptome was assembled using Trinity [47] on a CLC Genomic Server 8.0 with the following parameters: automatic word size, automatic bubble size and minimal length > 500 bp. The sequences obtained were blasted against the NCBI nt database and Open-Reading-Frames (ORFs) were predicted by scanning assembled sequences for ‘ATG’. We assessed the completeness of the de novo transcriptome with BUSCO [48].

### 2.3. Differential Gene Expression Analysis

We used Salmon software [49] for indexing, mapping and quantification of the transcripts for each of the LRT and SRT samples using the transcriptome assembly as a reference. We then used edgeR [50] for downstream analyses using standard procedures, which included (i) filtering the data, (ii) normalizing the data into effective library sizes [51] and (iii) estimation of the biological coefficient of variation (BCV). To identify significantly differentially expressed genes we use two approaches. First, we performed a gene-wise exact test for differences in the means of negative-binomially distributed counts between two conditions (LRT vs. SRT), with a *p*-value of 0.05 adjusted for false discovery rate (FDR) using the BH method [52]. We refer to the differentially expressed genes identified with this method as DEG set 1 (nine genes). Second, as none of the nine genes of DEG set 1 could be annotated for gene ontology analysis (see the section below), following other studies [25,53,54,55], we employed a more relaxed approach using a fixed threshold of *p*-value = 0.01 for the gene-wise exact test and without FDR correction. We refer to this second dataset as DEG set 2 (421 genes).

### 2.4. Gene Ontology Analysis

We assigned gene ontology (GO) profiles with FunctionAnnotator [56] using the GO category of behavioural processes (BP). We performed a gene set enrichment analysis (GSEA) to infer biological functions and pathways with topGO [57], using node size = 15 and ontology category = ‘BP’. For this analysis, we used the annotated genes with FunctionAnnotator from DEG set 2 (103 genes annotated of 421) as our group of differentially expressed genes, and our entire set of transcripts, which were annotated (14,794 genes) as our gene universe [57]. We used the Kolmogorov–Smirnov test, with a significance cut-off value of 0.01, because this is theoretically recommended over Fisher tests for pair-wise comparisons [58]. We generated a heatmap and performed a PCA with DEG set 2 to visualize differences in gene expression patterns among samples.

### 2.5. Statistical Analysis for Behavioural Data

We measured the time spent under shelters, number of visits to shelters and time delay until the first visit to a shelter, for each individual. To compare the time spent under shelters among the six samples (each composed of 10 LRT or SRT individuals) we used a Kruskal–Wallis test and Dunn post hoc test (with the BH method). The Mann–Whitney test was used to compare the number of visits and time delay until the first visit to shelter between SRT and LRT individuals. We used a Pearson correlation test to analyse the relationship between the mean resting time of each sample and the relative expression of genomic transcripts. Finally, we used the eigenvalues obtained from a PCA with all transcripts to explore the relationship between resting time and enriched GO terms. The significance threshold for these tests was kept at 0.05.

## 3. Results

### 3.1. Behavioural Experiments

At the end of the aggregation experiments, a majority of individuals were typically under one of the two shelters (9–12 cockroaches for all tested groups). There was a high inter-individual and inter-group variability regarding experimental resting time (Figure 1A), which is reflected in the high variability within and between samples (Figure 1B). This high variability of resting time across groups has been reported to be consistent over time, and to be a product of variation in the characteristics of the individuals who comprise each group [27]. Nevertheless, as expected, the total time spent under shelters was distinct between samples of LRT and SRT individuals (Kruskal–Wallis: H_5_ = 27.64, *p* < 0.001) and a Dunn post hoc test showed that LRT samples had longer sheltering times than SRT samples (Figure 1B, Appendix A). LRT individuals also visited the shelter earlier (Mann–Whitney: U = 271.5, *p* < 0.01; Appendix A) and more often (Mann–Whitney: U = 604.5, *p* = 0.02; Appendix A) than SRT individuals.

### 3.2. Differential Gene Expression

The transcriptome obtained from the total head RNA was comprised of 61,430 transcriptomic sequences and had a transcriptome-wide BUSCO completeness score of 72.4% (see Appendix A). The mean size of assembled transcripts was 1258 bp and the longest transcript was 18,670 bp. The total length of all transcripts was 77.3 Mbp. After normalizing and filtering, we retained 50,413 of these transcripts (see methods). BCV indicates that, in general, variability in expression values between replicates was relatively high (common dispersion = 0.06; Appendix A), meaning that expression values vary by ±24% between samples. When taking a look at how gene expression of each transcript was correlated to the mean time spent within shelters for each sample, we observe that the median (*r* = −0.01) and mean (*r* = −0.02) of Pearson correlation estimates were roughly symmetrical and very close to 0 (Figure 2A).

Our differential gene expression analysis with an FDR-corrected approach resulted in nine genes (DEG set 1). Of these nine DEGs (Appendix A), three were upregulated in LRT individuals while six were upregulated in SRT individuals. Only three of the nine genes were annotated by NCBI blast (see Appendix A), and these were identified as putative protein-coding genes with roles in odour-binding, amino-acid storage and phototransduction in *P. americana* (Table 1).

In contrast, with the relaxed cut-off used for DEG set 2, we obtained 421 differentially expressed genes (262 upregulated in SRT and 159 in LRT; see Appendix A for more details). These exhibited an inverse pattern of gene expression between LRT and SRT samples in several cases (Figure 2C), as reflected in the dendrogram, which clustered samples into LRT and SRT groups. The correlation analysis between the expression of genes in DEG set 2 and the average resting time of each sample (see Appendix A) generated a bimodal distribution of Pearson correlation estimates (*r*), indicating that the expression of these genes was either highly positively or highly negatively correlated with resting time. The PCA performed with DEG set 2 also separates SRT and LRT samples into distinct groups (Figure 2B). These results clearly separate LRT from SRT groups, despite the fact that pooling 10 individuals together per sample may obfuscate variation among individuals within pools and homogenize differences between pools, and that we use a relaxed cut-off *p*-value.

### 3.3. Gene Ontology Analysis

A total of 14,794 genes (of the dataset of 50,413 after filtering) were annotated and categorized into a total of 687 GO terms. As none of the nine genes in DEG set 1 were annotated, we used the annotated genes of DEG set 2 for the gene set enrichment analysis (103 genes annotated of 421 genes in DEG set 2). The GSEA indicated that 36 GO terms were significantly enriched and we retained 13 DEGs (from DEG set 2) enriched for these terms. The 36 GO terms could broadly be divided into three categories: muscle/post-larval development; ATP, glycolytic and fatty acid metabolic processes; axonogenesis (Appendix A). When considering all annotated genes involved in these GO terms, the SRT-upregulated genes were highly enriched for GO terms associated with these three categories (Figure 3A). On the other hand, LRT-upregulated genes were more relatively weakly enriched for these categories of GO terms (Figure 3A). When considering only the 13 DEGs enriched for these GO terms, the SRT-upregulated genes were enriched for all three categories of GO terms, while the LRT-upregulated gene was only enriched for the carbohydrate metabolic process (GO:0005975) (Figure 3B). It is also worth noting that these 13 genes have orthologs in taxa with solitary animals as well as highly integrated complex societies such as termites or ants (see Appendix A).

## 4. Discussion

Studies of organisms with rudimentary societies are useful for investigating the mechanistic basis of the early stages of group living, as they can help elucidate processes underlying the evolution of behavioural diversity, which is important for the effective performance of collective actions [65]. In this study, we show evidence of limited differential gene expression among cockroaches from two distinct behavioural phenotypes occurring during group aggregation dynamics, namely the short resting time (SRT) and long resting time (LRT) individual types (Figure 1A). We found nine differentially expressed genes between these types, with three of them identified as genes with putative roles in odour binding (PameOBP1), amino-acid storage (Cr-PI) and phototransduction (pTRP). In addition, our gene ontology and pathway analysis revealed 13 genes and several GO terms with functions in muscle development, glycolytic metabolism and axonogenesis; most of them enriched in SRT individuals.

In cockroaches, as in many other species, aggregation is a product of interactions between individuals [29,33,34]. The choice of shelter is mainly driven by chemical attraction between individuals based on hydrocarbons on the body surface and those passively deposited on the ground, which act as an aggregation pheromone and a social cue to indirectly estimate shelter quality [34]. We speculate below on how the three identified DEGs between LRT and SRT individuals might influence this process. The PameOBP1 protein is related to odour recognition (mates and food) and has been identified as a putative sexual pheromone receptor [59,60]. LRT individuals might thus have an enhanced ability (or lower threshold) to detect hydrocarbons, which could potentially explain their fast aggregation under shelters due to the strong retention effect promoted by hydrocarbons either passively deposited under shelters or on the body of congeners already sheltered [29].

The putative TRP (pTRP) protein, part of the TRPA family, was upregulated in SRT individuals. It has been suggested that this protein may play a role in light avoidance and photophobic behaviour in other species [62,63] as well as being responsible for the differentiation of phototransduction processes in cockroaches [64]. Cockroaches are photophobic and thus aggregate under shelters with dim light. Reaction to light stimuli (either in the form of negative phototaxis or through detection of shelters by the sudden decrease of light intensity upon entry) plays an important role in individual and group behaviour. As SRT cockroaches spend more time outside the shelter, and past research shows that the probability to visit a shelter is a consistent trait over time [37,40], pTRP could influence behaviour through moderating photophobia or light stimuli sensitization [64]. On the other hand, cockroaches have a high level of thigmotaxis, which can vary among individuals [38], and can play a role in aggregation and escape behaviour [66,67]. Thus, while none of the DEGs found could be related to thigmotaxis behaviour, differences in shelter use between LRT and SRT may be due to an interplay between phototaxis and thigmotaxis. Finally, the Cr-PI protein (upregulated in LRT individuals) is a part of the hexamerin protein complex [61,68]. This protein complex is involved in binding juvenile hormone [69], influences caste development in termites and honeybees [70,71] and has been implicated in amino-acid storage [61,72]. The accumulation of energy and amino acids that support the organism during non-feeding periods is an essential process in insects. Upregulation of Cr-PI could allow LRT individuals to spend longer non-feeding periods than SRT individuals. In support of this hypothesis, past research shows that less explorative (and less aggressive) cockroaches spend less time feeding compared to more aggressive/explorative individuals [73,74]. Further studies focusing on the role of hexamerins in cockroach behaviour could generate new insights into the formation of social dominance structures and help explain the transition from solitary to social life in Blattodea.

Our GO analysis indicated that genes involved in muscle development, metabolic/catabolic processes and axonogenesis were upregulated in SRT individuals. These results suggest a relationship between the SRT phenotype and aggressive or dominant behaviour, though we caution that these results are based on the use of an uncorrected significant threshold necessitated by a lack of annotation information. Previous studies across taxa show that bolder or more exploratory individuals have higher metabolic rates [4,75] and that axonogenesis-related terms may influence social interactions and social aggression [76]. Indeed, male hierarchy formation is a common trait in cockroaches, and our findings are supported by past studies with *P. americana* in seminatural conditions showing that more aggressive/dominant males were more mobile and spent more time outside (i.e., SRT type) [73,74,77]. However, agonistic behaviour and shelter occupancy in cockroaches can depend on the composition of the group [39,73]. Thus, while some of the candidate genes found in this study could provide a heritable mechanism for boldness or dominance as seen in other species [78,79], it is important to consider that environmental and social factors can also alter gene expression and thus moderate behaviour depending on context. Further analyses, preferably based on RNA samples of single individuals and using stricter controls or gene knock-down techniques [71], are needed to confirm the patterns we observed and provide more information on the heritable and environmental/social factors affecting behaviour and gene expression.

We find a limited number of differentially expressed genes and significantly enriched terms between SRT and LRT behavioural types. These results support the hypothesis that gregarious and subsocial species should be more totipotent and thus exhibit lower levels of differential gene expression between behavioural types than eusocial species, which we can expect to have accumulated additional changes associated with having more fixed social roles [23]. Additionally, there are several factors that may have influenced the outcome of the gene expression analysis. The subject used for this study is a non-model species, which means that fewer sequence records are available compared to model organisms such as honeybees or *Drosophila* sp. Additionally, the pool of 10 individuals may have introduced more complexity in terms of polymorphisms and alternative splicing, contributing to the fragmentation of transcripts [48,80]. These factors may have contributed to the relatively lower BUSCO completeness score and the lower percentage of GO annotations compared to similar studies with model insects.

## 5. Conclusions

In this study, we show that behavioural phenotypes with demonstrated effects on collective behaviour [27,37,39] correlate with differences in gene expression, though the limited number of differentially expressed genes, and even more limited number of identifiable genes, limits our ability to interpret these differences. Despite the technical shortcomings, we identified three genes differentially expressed between LRT and SRT samples, and we provide a list of 13 possible candidate genes (see GO analysis) affecting group dynamics. As similar phenotypes can have a shared molecular basis among species [71], these findings could help elucidate proximate mechanisms underlying animal personality and collective performance in other animal groups, though we caution that we have demonstrated patterns for only one species, and cockroaches themselves have diverse life histories which might influence these patterns. This diversity may be beneficial for evolutionary studies: cockroaches show a high diversity of life forms (from merely gregarious to social), reproductive strategies, dominance hierarchies and forms of cooperation, and ultimately represent the incipient stage of social evolution in the eusocial termites [26,33,71,81]. As such, cockroaches may prove a heuristic focal group for the study of the mechanistic basis of collective behaviour and by extension the evolution of sociality.

## Figures and Tables

**Figure 1 animals-12-02354-f001:**
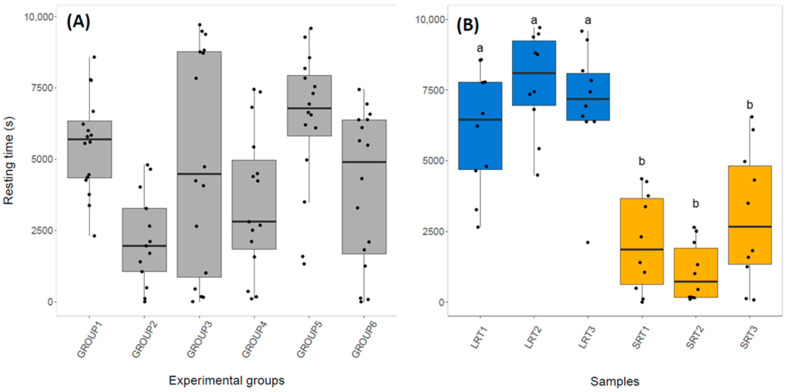
(**A**) Time spent under shelter for experimental groups and (**B**) for long resting time (LRT) and short resting time (SRT) individuals and groups. Letters indicate whether resting times differed significantly (a summary of Dunn test results can be seen in Appendix A).

**Figure 2 animals-12-02354-f002:**
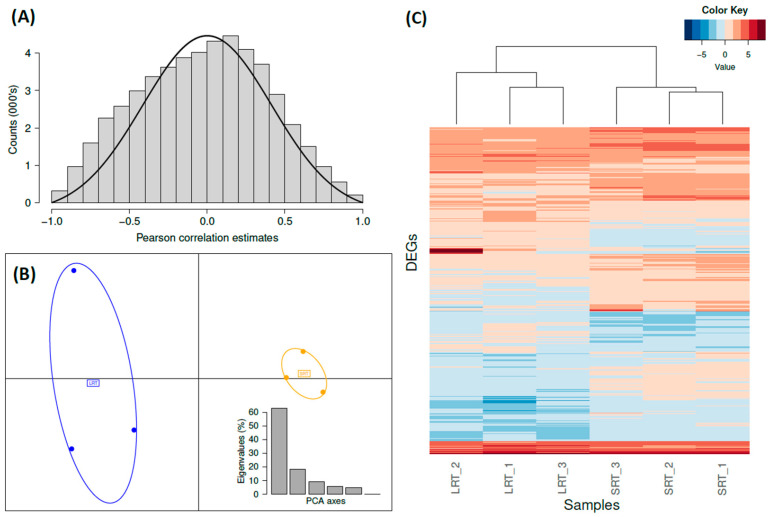
(**A**) Histogram of the Pearson correlation estimates (*r*) between CPM of all transcripts (50,413) and resting time of groups; the black line shows a normal distribution centred at 0 with the same SD as the observed distribution. (**B**) PCA (with 2 axes retained) with DEG set 2 and barplot with PCA eigenvalues for each axis. In blue are the LRT samples; in orange are the SRT samples. (**C**) Heatmap of 421 genes in DEG set 2 (logarithm of CPM).

**Figure 3 animals-12-02354-f003:**
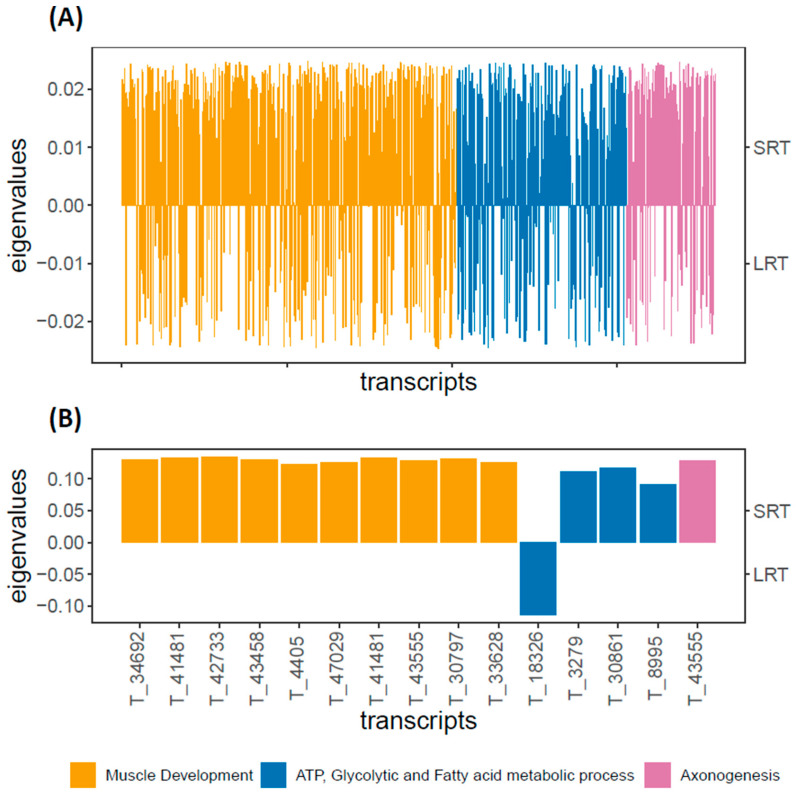
(**A**) Plot of PCA (performed with all 50,413 transcripts) eigenvalues (for PC axis 1) for all genes enriched for 36 GO terms (see Appendix A), and (**B**) the same but only eigenvalues for the 13 DEGs (from DEG set 2) enriched (see Appendix A).

**Table 1 animals-12-02354-t001:** Description of three DEGs identified and their putative functions.

Name of the Protein	Upregulated in	Function	References
PameOBP1	LRT	Odorant-binding protein; putative pheromone-binding protein; enriched expression in males	[59,60]
Cr-Pl/Per a 3	LRT	Arylphorin storage protein, part of hexamerins; storage of amino acids; bind juvenile hormone; melanin generation	[61]
pTRP	SRT	Major role in phototransduction	[62,63,64]

## Data Availability

All data to support the results of this article can be found in the Appendix A. Raw reads and assembled transcriptomes are available on the Sequence Read Archive (SRA; reference number: PRJNA864072).

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
