# Peer review of "Differential Gene Expression Correlates with Behavioural Polymorphism during Collective Behaviour in Cockroaches"

_animals, 2022, doi:10.3390/ani12182354_

Round 1

Reviewer 1 Report

The study provides a first step towards a better understanding of the molecular mechanisms associated with differences in boldness and behavioural plasticity in the american cockroach, a species that shows a high diversity of life forms (from merely gregarious to social), reproductive strategies, dominance hierarchies, forms of cooperation, and ultimately represent the incipient stage of social evolution.

The Authors nicely describe the aim of the study which is based on previous results. Experiments, Materials and Methods are nicely described, statistical analysis to my opinion correctly performed.

However, at least in my version of the manuscript, I do not find a Figure 2.

Minor point:

line 355: separate analysis and indicated

Author Response

Answer to Reviewer 1:

However, at least in my version of the manuscript, I do not find a Figure 2.

We are sorry that figure 2 was missing, there may have been some conversion error, or maybe an error during submission. We hope that it will be available now.

Minor point:

line 355: separate analysis and indicated

Corrected. See line 382.

Reviewer 2 Report

Overall, I find this to be a good and interesting paper. I enjoyed reviewing it. There are a few minor issues that need clarification, but nothing that cannot be easily addressed in a minor revision.

The introduction is good. It is concise, but sufficiently detailed. The methods related to subject preparation, behavioral experiments, and RNA extraction are all reasonable. The statistical methods also seem appropriate. I cannot speak in detail about methods related to gene expression; my laboratory is just starting to move in this direction. I hope the other reviewer(s) will be able to address this area in more detail.

There are some minor issues with the number of individuals collected from each group that weakens the paper slightly, but the authors are transparent in their methods.

The authors state that not using a shelter at all could be a sign of disease. Why then would reduced use of shelter in the SRT individuals not also be interpreted as a sign of some disease?

Why do the resting times differ so much across group (figure 1a)? Should we not expect that each group of 16 subjects has a similar overall resting time? I don’t see an issue with this variability in the scope of the authors’ work, but it should be addressed more.

There is a minor confound in use of shelters and assumption of photophobia. When using the shelter, the cockroaches may also be touching the support walls and top surface of the shelter with their antenna. For this reason, shelter use may also be related to thigmotaxis. A control procedure would include transparent shelters. There are a few studies investigating this particularly, using multiple transparent shelters with varying layers of red film, that could be cited in support of the authors’ interpretation. Some parts of the discussion should also address this potential confound. Due to previous work, this should not be a major issue, and I do not think anything needs to be removed from the paper. Rather a few comments to address this issue would be beneficial.

It might also be worth stressing that the authors’ conclusions are directed not toward the entire order of cockroaches, but toward P. americana, as there are over 4,000 cockroach species, with some being diurnal. Many may not share the tendencies of P. americana. In our work, for example, we have found that Eublaberus posticus startles more to the offset of light, than to the onset of light. In pilot research, we have found that, despite showing some photonegative tendencies in some tests, in other tests it has a clear preference for enclosed areas, even if they are well lit.

Author Response

Answer to Reviewer 2:

The authors state that not using a shelter at all could be a sign of disease. Why then would reduced use of shelter in the SRT individuals not also be interpreted as a sign of some disease?

There is a key difference between SRT individuals and those that spend zero time under shelters, and it is that SRT individuals still seek the presence of congeners, while in the extreme cases (~ zero time), individuals seem to start avoiding congeners, thus avoiding shelters. Additionally, while there is nothing strange with movement patterns of SRT individuals, individuals with zero times often show less mobility, walk slower around the arena, avoid presence of others, etc. There is no scientific record of this behaviour that we know of, although as a personal observation, to be on the safe side, we preferred to not use them. We added a statement to this effect in line 185.

Why do the resting times differ so much across group (figure 1a)? Should we not expect that each group of 16 subjects has a similar overall resting time? I don’t see an issue with this variability in the scope of the authors’ work, but it should be addressed more.

It is true that because of social amplification we often expect that all individual or groups behave very similarly, but not always. This variability at group level was reported in Planas-Sitjà et al 2015 – Proc Roy Soc B, where the variability across groups was consistent over time and could be explained by differences in group composition (now explained in lines 262-265).

There is a minor confound in use of shelters and assumption of photophobia. When using the shelter, the cockroaches may also be touching the support walls and top surface of the shelter with their antenna. For this reason, shelter use may also be related to thigmotaxis. A control procedure would include transparent shelters. There are a few studies investigating this particularly, using multiple transparent shelters with varying layers of red film, that could be cited in support of the authors’ interpretation. Some parts of the discussion should also address this potential confound. Due to previous work, this should not be a major issue, and I do not think anything needs to be removed from the paper. Rather a few comments to address this issue would be beneficial.

Indeed, thigmotaxis can be important in these cockroaches too. This is why we designed our shelters in order to reduce the thigmotaxis: 3 narrow legs instead of walls and height of 3 cm, so it can only be reached with antenna and not body. We added some lines about the thigmotaxis effect (see lines 157 & 365-369).

It might also be worth stressing that the authors’ conclusions are directed not toward the entire order of cockroaches, but toward P. americana, as there are over 4,000 cockroach species, with some being diurnal. Many may not share the tendencies of P. americana. In our work, for example, we have found that Eublaberus posticus startles more to the offset of light, than to the onset of light. In pilot research, we have found that, despite showing some photonegative tendencies in some tests, in other tests it has a clear preference for enclosed areas, even if they are well lit.

We added a clarification that our results only apply to our target species (see lines 424-427)